# Advancing Methodologies for Investigating PM_2.5_ Removal Using Green Wall System

**DOI:** 10.3390/plants13121633

**Published:** 2024-06-13

**Authors:** Claudia Falzone, Hugues Jupsin, Moussa El Jarroudi, Anne-Claude Romain

**Affiliations:** 1Research Unit SPHERE, Sensing of Atmospheres and Monitoring Laboratory, Department of Environmental Sciences and Management, University of Liege, 6700 Arlon, Belgium; acromain@uliege.be; 2Research Unit SPHERE, Eau—Environnement—Development Team, Department of Environmental Sciences and Management, University of Liege, 6700 Arlon, Belgium; h.jupsin@uliege.be

**Keywords:** climatic chamber, dry deposition, green wall, indoor air quality, PM_2.5_

## Abstract

Combustion processes are the primary source of fine particulate matter in indoor air. Since the 1970s, plants have been extensively studied for their potential to reduce indoor air pollution. Leaves can retain particles on their surfaces, influenced by factors such as wax content and the presence of hairs. This study introduces an innovative experimental approach using metal oxide particles in an office-like environment to evaluate the depolluting effect of plant walls. Two plant walls were installed in a controlled room, housing three plant species: *Aglaonema commutatum* ‘Silver Bay’, *Dracaena fragrans*, and *Epipremnum aureum*. Metal oxide particles were introduced via a compressed air blower positioned between the two walls. The concentration of these particles was monitored using PM_2.5_ sensors, and the deposition of iron (Fe) on the leaves was quantified through Inductively Coupled Plasma Mass Spectrometry (ICP-MS). This novel methodology effectively demonstrated the utility of both real-time sensors and ICP-MS in quantifying airborne particle concentrations and leaf deposition, respectively. The results revealed that *Dracaena fragrans* had a 44% higher Fe particle retention rate compared to the control (wallpaper). However, further validation through methodological replication is necessary to confirm the reproducibility of these findings.

## 1. Introduction

Pollution due to fine particle emissions is primarily an outdoor air quality concern, stemming from sources such as traffic, heating (wood, fuel, and coal), and industrial emissions [1,2]. However, indoor atmospheres can also harbor high concentrations of particles, from temporary sources or from external contaminants infiltrating through open windows [3]. Temporary sources, often linked to combustion processus like cooking [4,5], wood-burning stoves [6,7], candles [8], or cigarette smoke can contribute to indoor particle pollution, potentially leading to respiratory health issues (for example, asthma, chronic obstructive pulmonary disease (COPD) [9], or the development of lung cancer) [10].

Research since the 1970s has emphasized the role of plants in mitigating indoor air pollution. Plants possess natural air filtration abilities through various phytoremediation mechanisms including phytoextraction (phytoaccumulation), phytostabilization (phytoimmobilization), phytovolatilization, phytodegradation (phytotransformation), and phytofiltration [11]. Vegetation, particularly foliage in the case of green walls, can reduce the concentration of airborne particles by capturing them using their surfaces [12,13]. The effectiveness of particle capture depends on leaf morphology, with features like hairs and wax enhancing particles retention [14,15]. Rain and/or wind can remove particles from leaves, with larger particles washed away by rain and smaller ones encapsulated within leaf structure [16,17].

Experimental studies on particle deposition typically take place in the field, in wind tunnels or in a small chambre of various sizes. Field studies involve collecting leaves for analysis in the laboratory, which provides insights into the plant’s actual exposure to outdoor pollution [13,16,17,18]. Wind tunnel or small chambre experiments subject plants to an airflow containing artificially generated particles as example fluorescein sodium salt or ammonium sulphate or NaCl. This process is referred to as wet deposition. Alternatively, techniques such as generating particles through combustion phenomena or dispersing diamond powder into the air are classified as dry deposition methods [12,19,20,21,22,23,24].

Many laboratory studies employ the wet deposition technique; however, in the context of air pollution, particulate matter peaks are typically observed during dry weather conditions. Consequently, for research focused on particle deposition from air pollution, it is more advantageous to conduct laboratory experiments that closely replicate outdoor conditions.

In this study, the experiment was conducted in a closed, climate-controlled room equipped with two green walls, each measuring 140 cm by 210 cm. The particulate matter was simulated using 1 µm metal oxide microbeads. Fans were utilized to ensure the homogeneous distribution of particles throughout the air. The dry deposition on the leaves was subsequently quantified through precise laboratory analysis.

## 2. Materials and Methods

### 2.1. Experimental Chamber

The experiment was conducted within a meticulously controlled experimental climatic chamber, ensuring the precise regulation of temperature and humidity levels. The chamber dimensions, akin to those of a typical office space, comprised a floor area of 20 m^2^ and a volume of 60 m^3^. Encircling the chamber was a buffer zone, thermally regulated to facilitate subjecting the climatic chamber to diverse, reproducible temperature profiles. Thirty-nine sensors were strategically positioned within the experimental area: 37 for temperature measurement (comprising air temperature, resultant temperature, and surface temperature), 1 for relative humidity, and 1 for CO_2_ monitoring. Figure 1 provides a schematic representation of the chamber layout.

To achieve air homogenization throughout the experiment, two fans were deployed. Natural air renewal during the test, reflecting a value of 0.044 h^−1^, mimicked the natural infiltration process. Four distinct devices were employed to measure PM_2.5_ levels within both the chamber and the buffer area (refer to Table 1 for details). Device No. 1, a custom-made apparatus equipped with a Sensirion^®^ sensor SP30, and Device No. 2, a commercial instrument (TSI QUEST^®^ Technologies EVM Series Environmental Monitor System), were positioned on a table at the chamber’s center, offering continuous measurement capabilities alongside periodic weighing. Additionally, a third device (No. 3), developed and provided by Comon Invent^®^, was affixed to the chamber ceiling. In the buffer area, a second custom-made device (No. 4, Sensirion^®^) was installed. Table 1 outlines the specifications of these sensor devices. Throughout the test, the average temperature and relative humidity were maintained at 20 °C and 70%, respectively.

### 2.2. Green Wall

Two green walls, with a size of 140 cm length and a height of 210 cm, were fixed to the chamber walls (Figure 2). The sphagnum moss substrate was contained in galvanized steel baskets. Each green wall was composed of ten different species of plants known for their depolluting effect on indoor air: *Aglaonema commutatum* ‘silver bay’, *Epipremnum aureum*, *Nephrolepis exaltata* ‘Bostoniensis’, *Dracaena fragrans*, *Chamaedorea elegans*, *Spathiphyllum wallisii* ‘sensation’, *Chlorophytum comosum* ‘Ocean’, *Hedera helix* ‘Pittsburgh’, *Begonia rex* ‘Alaska creek’, and *Tradescantia zebrina*.

### 2.3. Particulates

The challenge was to find particles that we could diffuse in the room, which were of the recommended size, and above all did not present any risk in terms of toxicity. It was in fact unthinkable to contaminate the study room with heavy metals such as cadmium or lead, even if these would have allowed for a lower limit of quantification when measuring the part adsorbed by the leaves. After consultations with the occupational health organism, it was found that the iron oxide particles met the criteria for toxicity, particles size, and ease of dosing. Iron oxide (Fe_3_O_4_) particles were used in this study (Nanography^®^, Purity: 99.5+%, Size: 1 μm).

Diffusion into the room was achieved using a spray system developed as part of this study (Figure 3). It consists of a funnel, the lower part of which is blocked by cotton, with a compressed air hose connected to the lower part. One gram of iron oxide powder was placed in the funnel before a sudden charge of compressed air was sent out, ejecting the particles into the air in the room. The presence of two fans in the room helped to homogenize the particles in the air.

### 2.4. Sampling of PM_2.5_

#### 2.4.1. Ambient Indoor Air

The collection of suspended particles was conducted using TSI’s QUEST instrument for 36 h and 13 min (time step 5 s, n = 26,077). The device is equipped with a pump designed to capture particulate matter on a filter. The filter is circular in shape with a diameter of 34 mm for an empty weight of 0.34601 g. A real-time measurement was also conducted using the devices described in Section 2.1.

#### 2.4.2. Leaves and Reference Materials

Three samples of different types of leaves were chosen, each with a sufficiently large surface area to provide a useful signal for iron determination. The three plants chosen for their different shape of leaf types are the following species: *Aglaonema commutatum* ‘silver bay’, *Dracaena fragrans*, and *Epipremnum aureum* (Figure 4). A sample of the substrate was also took, i.e., sphagnum moss from Madagascar, to check whether it was also involved in particle adsorption. The leaves and sphagnum moss were taken from the left wall, about 20 cm from the edge, to ensure central sampling in relation to the two panels.

In order to determine the adsorption of a wall covering in relation to that of the plants, two types of wallpaper were used as controls. They differ in their structural characteristics: the first has a structured appearance with numerous asperities, while the other is much smoother (Figure 5). During the experiment, the control samples, made up of two types of wallpaper, were placed halfway up the green walls: on the left, in a central position (between the two walls), and to the right of the two green walls (stars in Figure 1).

A reference sample was taken before the particles were introduced, so iron analyses were carried out on the three types of leaf, on the sphagnum moss, and on the two types of wallpaper.

In order to standardize the measurements, the adsorption must be reduced to the unit of surface area, whether for plants or wallpaper, and to the unit of weight for sphagnum moss. The public domain ImageJ software created by Wayne Rasband of the National Institutes of Health in the United States was used (version 1.54d). This software is used to determine the surface area of the leaves by comparing them with a reference surface on the same image (77 mm by 77 mm post-it note). The photo must be taken vertically in relation to the surfaces (Figure 6). The sheet was previously covered with a pane of glass to flatten it so that its surface could be precisely determined.

The wallpaper samples fixed to the wall of the room all had the same dimensions, 150 mm by 96 mm. The quantity of sphagnum moss was determined by weighing the fresh weight before analysis and the dry weight after iron desorption.

#### 2.4.3. Analysis Laboratory—Inductively Coupled Plasma Mass Spectrometry (ICP-MS)

Measurement of the iron particles adsorbed on the leaves was carried out by Inductively Coupled Plasma Mass Spectrometry (ICP-MS). The particles present on the leaves and pieces of wallpaper were recovered and solubilized using a 1% HNO_3_ solution. This procedure was dispensed with the complete mineralization step usually employed for ICPMS assays. The blanks and samples taken after iron dispersion underwent the same procedure: rinsing with HNO_3_ solution and recovery of the eluate in a 100 mL beaker and making up to the mark.

ICP-MS is a powerful analytical technique used for the precise determination of metal concentrations, with, in this case, a particular focus on iron (Fe). In this method, a sample is nebulized and introduced into an argon plasma, where it undergoes ionization. The resulting ions are then separated based on their mass-to-charge ratios, and the detector quantifies their abundance (Figure 7) [27,28].

For iron analysis by ICP-MS, a standard calibration curve is established using known concentrations of iron standards. The instrument’s sensitivity and response are optimized for iron detection through the adjustment of parameters like the radiofrequency power and nebulizer gas flow. Isotopes of iron, such as Fe_56_ and Fe_57_, are commonly monitored for accurate quantification; in this study, we chose Fe_56_ [28,29].

Sample preparation is crucial in iron analysis, involving meticulous digestion methods to ensure complete solubilization of the metal, but in this case, the HNO_3_ 1% solution was used to avoid the digestion process. Internal standards are employed to correct for matrix effects and enhance accuracy [30]. The obtained data can be further processed using specialized machine software to calculate the concentration of iron in the original sample.

A quadrupole Thermo iCAP RQ ICPMS equipped with a collision cell and an automatic sampler for iron dosing was used. The calibration of Fe_56_ was carried out between 0 and 5000 ppb, the limit of detection (LOD) was 1.374 ppb, the background equivalent concentration (BEC) was 1.234 ppb, and the relative standard error (RSE) was 3.82%.

## 3. Results

### 3.1. Airborne PM_2.5_

The continuous measurements of PM_2.5_ concentrations in the chamber reveal a sharp rise in concentration after the ejection of particles, immediately followed by a gradual decrease, exhibiting an inverse logarithmic trend (Figure 8a–c corresponding to devices 1, 2, and 3, respectively).

The Comon Invent device (device n°1—Figure 8a) suspended from the ceiling shows a concentration peak higher than the concentration measured by the sensor places on the table in the room. However, this peak decreases more quickly than the other peaks before reaching similar concentrations measured by the other sensors.

The sensor located in the buffer area (device n°4—Figure 8d) shows a similar profile, but the concentration values are significantly lower than the concentration measured inside the climate chamber. In addition, the increase in the particles’ concentration shows a positive lag compared with the increase in the concentration in the climate chamber, confirming that this slight increase is linked to the natural exchange of air between the chamber and the buffer zones.

Before the experimental test, the particles’ concentration measured in the room was 0 for all sensors. The concentration began to increase directly after the particles were emitted at 10:47 a.m. (27 October 2023). The maximum concentration was reached at 10:47 a.m., 11:08 a.m., and 10:58 a.m. for sensors 1, 2, and 3, respectively, with values of 377 µg/m^3^, 134 µg/m^3^, and 148 µg/m^3^, respectively. Sensor 4 (buffer zone) reached its maximum concentration at 01:14 p.m., with only 2 µg/m^3^. Sensor 1 showed a similar concentration range and decreasing concentration profile to the other sensors located in the chamber from 11:00 onwards, with a starting concentration of 178 µg/m^3^. The concentration values at 00:00 a.m. (28 October 2023) were 12, 8, 10, and 0 for sensors 1, 2, 3, and 4, respectively.

### 3.2. PM_2.5_ Sucked onto the Filter

The quantity of particles present in the air sampled on the filter was 0.588 mg (0.352–0.346 g). This value includes the metal oxide particles, as well as other coarser particles that may be present in the air. In order to obtain the exact dosage of metal oxide, a laboratory analysis by ICPMS was carried out. The mass of metal oxide present on the filter was 0.139 g/m^2^ for filter sampling, lasting 36 h and 13 min.

### 3.3. PM_2.5_ Dry Deposition on the Leaves

The surfaces of the leaves were measured using ImageJ (v1.46 R) software and compared with a known surface. Table 2 shows the surfaces measured for the leaves and wallpaper (i. 1 to 4) and the weight of the sphagnum moss collected (i. 5).

Table 2 may seem confusing, so it is useful to recall the procedure developed: the leaf from each species was cut, which was then measured and rinsed with water and acid in the laboratory, so there was a leaf for the blank test “unexposed” and a sheet for the test. The same goes for wallpaper. This procedure explains the different surfaces for leaves. As the sphagnum surface cannot be measured using optical techniques, the only solution was to relate the adsorption to the dry weight.

The concentrations measured for the plant and sphagnum samples are presented in Table 3. For each sample, these concentrations were converted into grams of iron per unit area.

Sphagnum moss is particularly hydrophilic, which is why it was used as a support for the plants. In order to make the results as comparable as possible, the adsorption was expressed as kg dry weight. The percentage of dry matter was determined by drying at 105 °C to constant weight, and an average of 4.8% was recorded. In this way, a value of 1.6 g Fe/kg dry moss was obtained.

Table 4 shows the Fe concentrations measured by ICP-MS for the control wallpaper samples. As for leaves, these concentrations were converted into grams of iron per unit area. The concentration of Fe varies according to their location.

The mass per unit area values calculated for each plant and wallpaper sample were compared with the reference value. The choice of this reference value corresponded to that of the smooth wallpaper positioned between the two green walls (i = 2 in Table 4). The smooth texture of the leaves and the proximity of this wallpaper to the sampling zone guided the choice of this value. Consequently, the value of 0.74 g Fe/m^2^ will be used as a reference. The percentage of Fe deposition was calculated to identify which plants are best at retaining iron oxide particles on their leaves (Table 5).

## 4. Discussion

The selection of metal oxide microparticles in this study was primarily driven by the analytical advantages provided by ICP-MS, which allows for the precise quantification of elements at very low concentrations. Iron was chosen over other metals such as cobalt, cadmium, and lead due to its relatively low toxicity, making it safer for experimental purposes [13,31]. A particle size of 1 µm was selected to effectively represent typical particulate matter pollution in the air, aligning with the sizes that most affect air quality and human health [32].

The study successfully demonstrated that these iron oxide microbeads can be measured in the air using PM_2.5_ sensors and on surfaces post-deposition using ICP-MS. The PM_2.5_ sensors were indeed sensitive enough to detect the 1 µm particles. However, the microbead samples exhibited a tendency to agglomerate due to magnetic interactions, requiring dispersion with compressed air to achieve finer particle agglomerates [15]. Despite some residue remaining in the cone due to electrostatic interactions, the quantity of particles emitted into the air was sufficient for the experiment, and the particle size allowed them to remain suspended for several hours, ensuring ample time for interaction and deposition on the experimental surfaces.

PM_2.5_ measurements showed a consistent temporal pattern with a sharp initial increase in concentrations followed by a gradual decrease, described by logarithmic decay. Sensor 1, suspended from the ceiling, recorded higher peak concentrations and an earlier increase compared to sensors 2 and 3, which were placed on a table. This was anticipated as the particles were dispersed from the middle of the room, causing the upper part of the room to receive a concentrated particle sample before they were mixed throughout the room by fans. Over time, the concentration measured by all sensors equilibrated, indicating the effective mixing of the particles. Sensor 4, located outside the chamber, showed negligible concentrations, confirming that the chamber was adequately isolated from the external environment [31].

Despite the apparent homogenization of particles in the air, ICP-MS measurements of Fe on control samples revealed a non-uniform distribution of metal oxide particles. Specifically, controls on the right side of the chamber had significantly lower Fe concentrations than those on the left, likely due to aerodynamic effects within the room. The wall selected for plant sampling was well exposed to the ambient air concentration of metal oxide particles.

Comparing smooth and rough control wallpapers revealed an unexpected result: smooth wallpaper retained 10–30% more particles in the most exposed areas, contrary to the hypothesis that rougher surfaces would retain more particles. This discrepancy may stem from differences in the fiber structure of the wallpapers or the impact of pigmentation on surface roughness [33]. Further research should explore particle retention across various substrates and plant species to elucidate these mechanisms.

Among the three plant species analyzed, *Dracaena fragrans* and *Aglaonema commutatum* exhibited higher retention rates of Fe (+44% and +3%, respectively) compared to the control, while *Epipremnum aureum* had a lower retention rate (−17%). The green wall design incorporating sphagnum moss showed that the moss contributed slightly to particle deposition (0.08 g Fe/kg), primarily through its exposed parts. Plants in front of the moss acted as filters, enhancing particle deposition. Active filtering using a pump collected significantly fewer particles than passive deposition on control surfaces, underscoring the superior efficiency of surface interactions over forced airflow for capturing suspended particles [13].

## 5. Conclusions

The novel methodology utilizing metal oxide particles presents a promising approach for studying particulate deposition on surfaces. Our initial findings demonstrate the efficacy of real-time sensors and ICP-MS laboratory analysis in quantifying particulate concentrations both in the air and on surfaces. *Dracaena fragrans*, in particular, emerged as a highly effective candidate for indoor air particulate decontamination, exhibiting a 44% higher particle retention rate compared to the control.

However, these preliminary results necessitate further validation through method replication to ensure the reproducibility of the experiment. To strengthen our conclusions, it will be essential to replicate measurements across multiple leaves of the selected plant species. Additionally, broadening the scope of this investigation to include other plant species could provide a more comprehensive understanding of phytodepuration efficacy and identify new avenues for research.

In conclusion, while these preliminary findings lay the groundwork for significant advancements in our understanding of indoor air phytodepuration, they require thorough validation and continuous exploration to fully realize their potential. Expanding this research could ultimately lead to more effective strategies for mitigating indoor air pollution through the use of plant-based systems.

## Figures and Tables

**Figure 1 plants-13-01633-f001:**
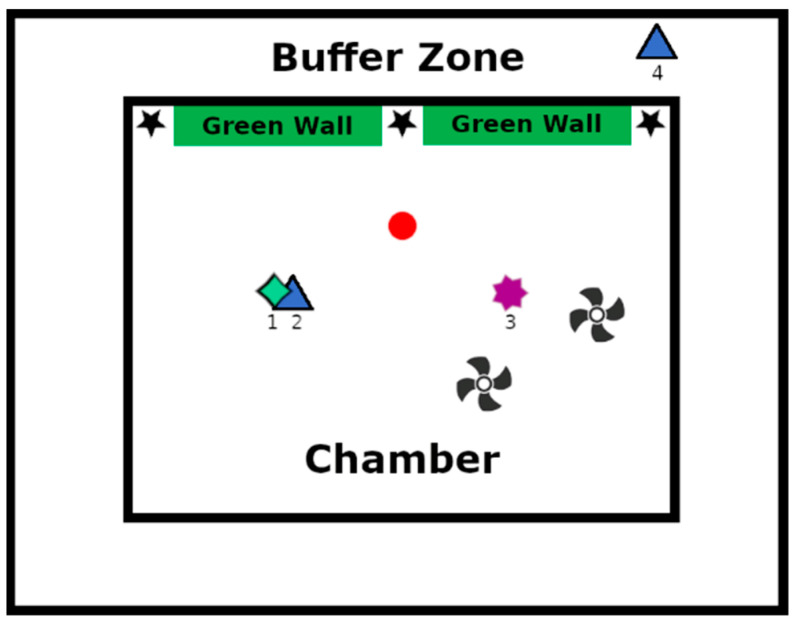
Schematic view of the climatic chamber and the buffer area. Stars = reference material (two different wallpapers). PM_2.5_ sensors: (1) Comon Invent^®^ device suspended from ceiling, (2) Sensirion^®^ Sensor SP30 on a table, (3) TSI Quest^®^ device on a table, (4) Sensirion^®^ Sensor SP30 on a table. Circle = emission of particulates. Fan logo = fan.

**Figure 2 plants-13-01633-f002:**
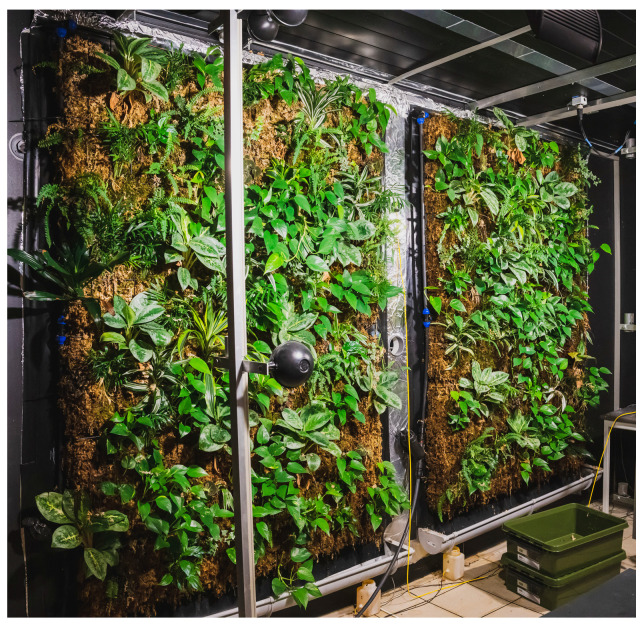
Two green walls each with a dimension of 140 cm × 210 cm; substrate—sphagnum moss; plants—mix of 10 species.

**Figure 3 plants-13-01633-f003:**
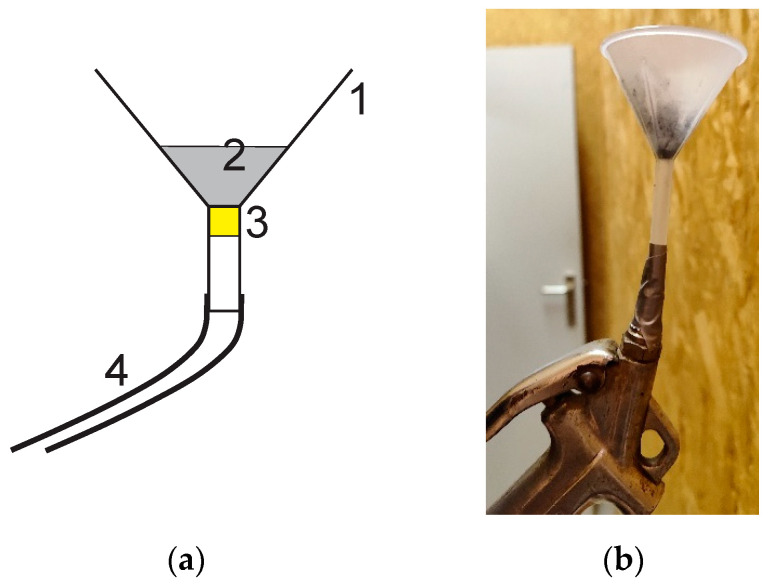
(**a**) Scheme of the particle spraying system: 1. funnel, 2. iron oxide, 3. cotton wool, 4. compressed air hose; (**b**) Photo of system.

**Figure 4 plants-13-01633-f004:**
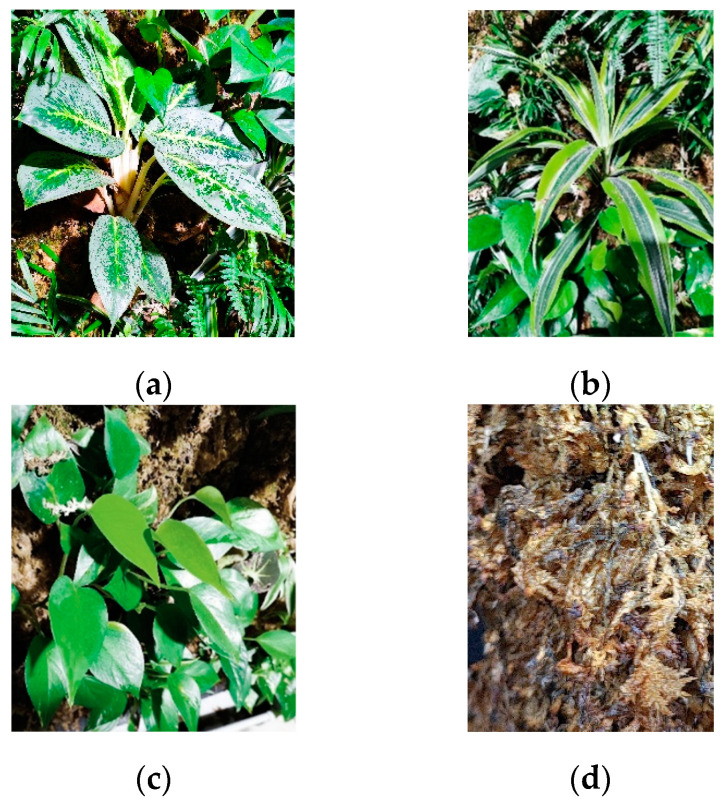
Plants and sphagnum moss used for the experimentation: (**a**) *Aglaonema commutatum* ‘silver bay’, (**b**) *Dracaena fragrans*, (**c**) *Epipremnum aureum*, and (**d**) sphagnum moss.

**Figure 5 plants-13-01633-f005:**
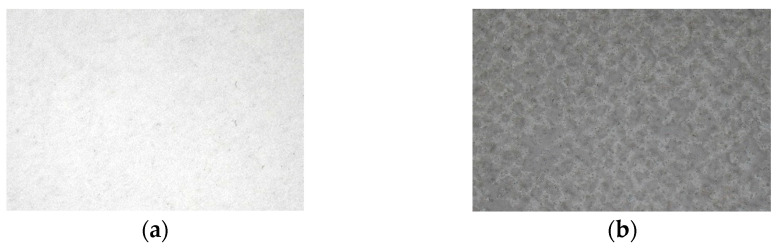
Reference material for the experimentation: (**a**) smooth wallpaper and (**b**) structured wallpaper.

**Figure 6 plants-13-01633-f006:**
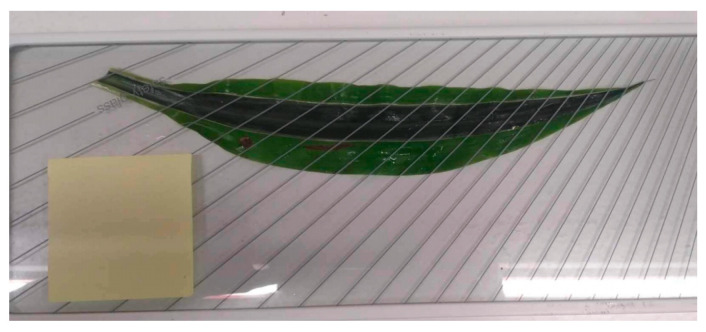
Measurement of sheet surface area (Dracaena fragrans) using a control unit (77 mm by 77 mm post-it note).

**Figure 7 plants-13-01633-f007:**
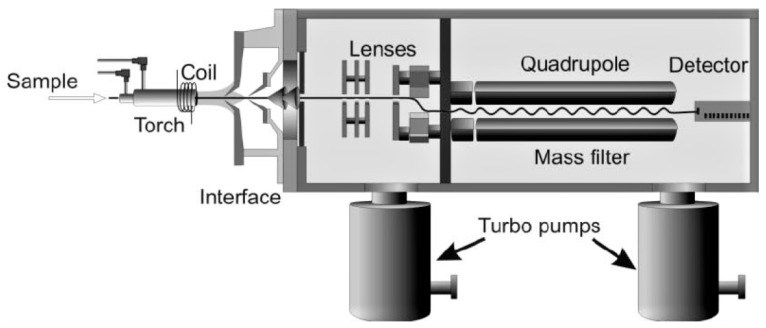
Cross-section schematic of an ICP-MS.

**Figure 8 plants-13-01633-f008:**
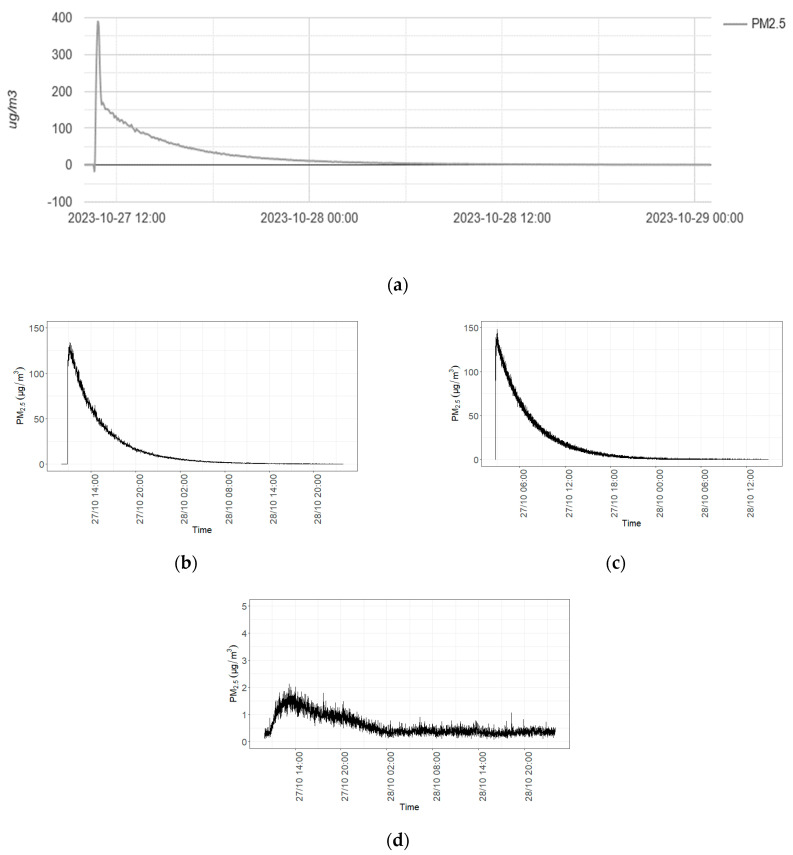
PM_2.5_ concentrations from the sensors: (**a**) Comon Invent suspended from the ceiling in the chamber (1), (**b**) Sensirion SP30 (2) placed on the table in the chamber, (**c**) QUEST (3) placed on the table in the chamber, and (**d**) Sensirion SP30 (4) placed on the table in the buffer.

**Table 1 plants-13-01633-t001:** Summary of sensors used.

Sensor’s Brand	Technology	Range	Resolution	Accuracy
Sensirion SP30 [25]	Laser-based scattering	0 to 1000 µg/m^3^	1 µg/m^3^	@25 °C ±10 µg/m^3^ ±10% at 100 to 1000 µg/m^3^
Quest EVM Series [26]	Nephelometry and gravimetric sampling	0 to 200 mg/m^3^	No data	0.001 mg/m^3^
Comon Invent^®^	No data	No data	No data	No data

**Table 2 plants-13-01633-t002:** Leaves (area), wallpaper (area), and sphagnum moss (weight) measurements. The unexposed surface corresponds to a sample before exposition and the exposed surface to a sample after exposition.

i	Denomination	Unexposed Surface (cm^2^)	Exposed Surface (cm^2^)
1	*Aglaonema commutatum* ‘silver bay’	79.21	53.25
2	*Epipremnum aureum*	55.32	47.26
3	*Dracaena fragrans*	99.95	95.63
4	Wallpaper control	144.00	144.00
		Gross weight (g)	
5	Sphagnum moss	13.40	31.34

**Table 3 plants-13-01633-t003:** Fe concentrations (in 100 mL) for unexposed and exposed samples. The g Fe/m^2^ values were calculated on the basis of the concentrations. The difference corresponds to quantity of PM deposition on leaves or sphagnum moss (exposed–unexposed).

i	Denomination	[Fe] Unexposed (ppb)	[Fe] Exposed (ppb)	Unexposed (g Fe/m^2^)	Exposed (g Fe/m^2^)	Deposition of Fe (g Fe/m^2^)
1	*Aglaonema commutatum* ‘silver bay’	13,066	12,839	1.65	2.41	0.76
2	*Epipremnum aureum*	3399	5786	0.61	1.22	0.61
3	*Dracaena fragrans*	7828	17,659	0.78	1.85	1.06
					(g Fe/kg)	
4	Sphagnum moss	1567	6139	0.12	0.2	0.08

**Table 4 plants-13-01633-t004:** Fe concentrations (in 100 mL) for exposed controls. The g Fe/m^2^ values were calculated on the basis of the concentrations.

i	Type of Wallpaper	Location	[Fe] Exposed (ppb)	Exposed (g Fe/m^2^)
1	Smooth	Star left in Figure 1	12,769	0.89
2	Smooth	Star center in Figure 1	10,617	0.74
3	Smooth	Star right in Figure 1	157	0.01
4	Structured	Star left in Figure 1	11,634	0.81
5	Structured	Star center in Figure 1	8215	0.57
6	Structured	Star right in Figure 1	256	0.02

**Table 5 plants-13-01633-t005:** Percentage of Fe deposition on leaves compared to the reference value i = 0.

i	Denomination	Deposition of Fe (g Fe/m^2^)	Difference (%)
0	Smooth center wallpaper	0.74	0
1	*Aglaonema commutatum* ‘silver bay’	0.76	3
2	*Epipremnum aureum*	0.61	−17
3	*Dracaena fragrans*	1.06	44

## Data Availability

All the data are presented in this paper.

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
