# Peer review of "Advancing Methodologies for Investigating PM2.5 Removal Using Green Wall System"

_plants, 2024, doi:10.3390/plants13121633_

Round 1
Reviewer 1 Report
Comments and Suggestions for Authors
Title: Advancing Methodologies for Investigating PM2.5 Removal Using Green Wall System
This paper describes the application of the Green Wall System to remove PM2.5 and the ability to retain Fe. It uses PM2.5 sensors to measure the indoor level of PM2.5 and demonstrates the quantification of Fe using ICP-MS. The study is innovative in approach, and will certainly catalyze new research in the application of the Green Wall System to remove indoor air pollutants. The paper is concise and well-structured, although there are some weaknesses in written style and presentation that need to be cleaned up. Overall the paper is appropriate for publication in Plants once the existing deficiencies have been addressed.
- The underlying mechanism for the dry deposition of Fe. The paper describes the differences in the dry deposition of Fe across plant species in the green wall system. The author highlights the role of wax content and the presence of hairs in leaves for PM2.5 removal in the abstract. Do the wax content and presence of hairs in leaves work for the dry deposition of Fe? If yes, the authors should add some experiments to prove the interactions between Fe and wax content or the presence of hairs in leaves. If not, what are the main factors driving the differences in the dry deposition of Fe across plant species in the green wall system? The authors discuss the underlying adsorption mechanism.
- Plant physiology. Plants can emit metal-containing nanoparticles through leaves. What do the emissions of metal-containing nanoparticles from plants on the deposition of Fe in the Green Wall System?
Zhang, W., Cao, J., Luo, X. et al. Plants emit sulfate-, phosphate- and metal-containing nanoparticles. Environ Chem Lett 21, 655–661 (2023). https://doi.org/10.1007/s10311-023-01567-5
Author Response
See the appendix.

Reviewer 2 Report
Comments and Suggestions for Authors
Keywords
Rearrange the keywords alphabetically.
Introduction
Please mention the study gap at the end of the first paragraph.
The third paragraph is very short and incomplete; please enrich and complete it.
Results
A few figures can be supplementary.
In Figure 8, PM2.5 ‘2.5’should be subscript.
Discussion
The discussion section is currently weak despite having many results. Briefly restate the key findings from the results. Analyze the significance of these findings, explain how they contribute to the existing knowledge, compare them with previous studies, and highlight any patterns or trends. Discuss potential reasons behind the observed results and their implications.
Consider any limitations and suggest areas for future research.
Conclusion
Please write one logical paragraph.
Comments on the Quality of English LanguageMinor editing of English language required
Author Response
See the appendix.

Round 2
Reviewer 1 Report
Comments and Suggestions for Authors
accept
Reviewer 2 Report
Comments and Suggestions for Authors
Accept in present form
Comments on the Quality of English LanguageMinor editing of English language required